behaviour

bird hearing, penguins, bioacoustics, playback

**Author for correspondence:**
K. Sørensen
e-mail: kenneths@biology.sdu.dk

# Gentoo penguins (*Pygoscelis papua*) react to underwater sounds

K. Sørensen[1], C. Neumann[1], M. Dähne[2], K. A. Hansen[1] and M. Wahlberg[1]

[1]Department of Biology, University of Southern Denmark, Campusvej 55, 5230 Odense M, Denmark
[2]German Oceanographic Museum Foundation, Katharinenberg 14-20, 18439 Stralsund, Germany

KS, 0000-0002-1256-2495

Marine mammals and diving birds face several physiological challenges under water, affecting their thermoregulation and locomotion as well as their sensory systems. Therefore, marine mammals have modified ears for improved underwater hearing. Underwater hearing in birds has been studied in a few species, but for the record-holding divers, such as penguins, there are no detailed data. We played underwater noise bursts to gentoo penguins (*Pygoscelis papua*) in a large tank at received sound pressure levels between 100 and 120 dB re 1 µPa RMS. The penguins showed a graded reaction to the noise bursts, ranging from no reactions at 100 dB to strong reactions in more than 60% of the playbacks at 120 dB re 1 µPa. The responses were always directed away from the sound source. The fact that penguins can detect and react to underwater stimuli may indicate that they make use of sound stimuli for orientation and prey detection during dives. Further, it suggests that penguins may be sensitive to anthropogenic noise, like many species of marine mammals.

## 1. Introduction

Marine mammals and diving birds are secondarily adapted to the aquatic environment. Their anatomy and physiology, initially tuned for terrestrial life, have been modified to also function in water. Some species can dive to depths of more than 500 m for more than an hour in pursuit of their prey, which include fish, cephalopods and crustaceans. Their bodies are streamlined to reduce drag and preserve body heat while diving. Indeed, many species hunt for fish year-around in cold waters while keeping their core body temperature intact [1].

In addition to these adaptations, some of the sensory systems of marine mammals and diving birds have been adjusted to

function well not only in air but also under water. In air, most mammals and birds have acute visual and hearing abilities, and some also rely to a large part on olfaction. When submerged, these senses function differently due to different properties of water and air as propagation media for light, sound and chemical stimuli. For example, the difference in the refraction index between air and water challenges the eyes' abilities to focus the image on the retina [2]. The morphology of the eyes of whales and seals have been adapted to function well both in air and under water [2,3]. Some marine birds, such as penguins, also seem to have anatomical adaptations to make their eyes function well in both media [4].

Besides vision, the sense of hearing of an air-adapted animal is also challenged under water. The speed of sound is almost 4.5 times faster in water than in air. Together with the 800 times higher density of water, this calls for anatomical changes in the detection system for it to function optimally [5]. The high underwater speed of sound also challenges the animals' abilities to determine the direction to a sound source. In air, directional hearing in the horizontal plane is accomplished by measuring the time lag and intensity difference between the sound pulse received at the two ears. Under water, neither of these cues function very well, as the higher speed of sound results in shorter time lags as well as longer wavelengths, and therefore smaller intensity differences between the two ears [5,6].

If the difficulties with hearing under water can be overcome, aquatic sound is an excellent sensory channel due to the high speed of sound and very efficient sound propagation in water. Hearing is of special importance in the aquatic environment, where sound can be transmitted over considerably larger ranges than light. Marine mammals have modified ears to allow for not only low hearing thresholds under water and a large receiving bandwidth, but also excellent abilities to tell the direction to the sound source. Among marine mammals, we find some of the most acute hearing abilities of any animal, both in terms of receiver bandwidth and sensitivity [7]. Whales, spending all their lives in water, have tuned their hearing to work better under water than in air, whereas seals, spending time both on land and in water, have excellent hearing abilities both above and below the water surface [8–10].

Besides whales and seals, little is known about hearing adaptations to the aquatic environment in secondarily adapted aquatic vertebrates. Sea otters (*Enhydra lutris*) have relatively low hearing thresholds in a broad frequency range in both media [11], and manatees (*Trichechus manatus)* hear comparably well under water [12]. Some aquatic turtles (e.g. *Trachemys scripta*) have tuned their hearing to function in both media, even though the sensitivity and frequency range is restricted compared to marine mammals [13]. Polar bears (*Ursus maritimus*) have hearing abilities comparable to other similar-sized mammals on land [14], but their hearing abilities under water have not been studied.

Much less is known about underwater hearing in marine birds. In air, all avian species, including marine birds such as penguins, have excellent hearing abilities [6,15–17]. They use airborne sounds for behaviours such as mating displays, courtship, predator warning and to establish the relation between mother and chicks [18]. The extremely elaborate sound communication signals used by many species indicate not only the importance of hearing to birds, but also their highly acute abilities to detect and differentiate between different sounds and sources. For most birds, the frequency range and sensitivity of hearing are poorer than in mammals [6]. Their directional hearing abilities are surprisingly good, also at lower frequencies, which is explained by the acoustic coupling of the two ears [19].

Underwater hearing has so far been studied in two species of marine birds, the long-tailed duck (*Clangula hyemalis*) and the great cormorant (*Phalacrocorax carbo*), both of which detect underwater sound [20,21]. Many marine birds actively chase prey such as fish during longer or shorter dives. Some of the most extreme species are penguins, which are highly specialized for an aquatic lifestyle, and dive during travelling and foraging. For some penguins, foraging dives can be as deep and long as in some species of seals at mesopelagic depths [22]. Being visually oriented predators, penguins have good visual abilities and fully functional eyes under water, which are used for hunting [23–25]. Still, given the excellent acoustic properties of water, one may wonder if penguins do not also make use of auditory cues under water while diving.

Understanding the underwater hearing abilities of marine birds is also important for conservation efforts. If penguins hear well under water, they may be affected by anthropogenic noise. The acute hearing abilities of marine mammals make them vulnerable to human-induced sounds from shipping, sonars and air guns [20,26,27]. Marine mammals have been shown to interrupt their feeding activities due to ship noise [28]. For louder sound sources (e.g. pile-driving noise), marine mammals can be displaced by tens of kilometres [29]. The most extreme effects of anthropogenic sounds concern beaked whales, which may strand due to naval sonar signals produced during military exercises [30]. If marine birds are also affected by anthropogenic noise, this needs to be considered in efforts to mitigate the effects of noise emissions in the marine environment.

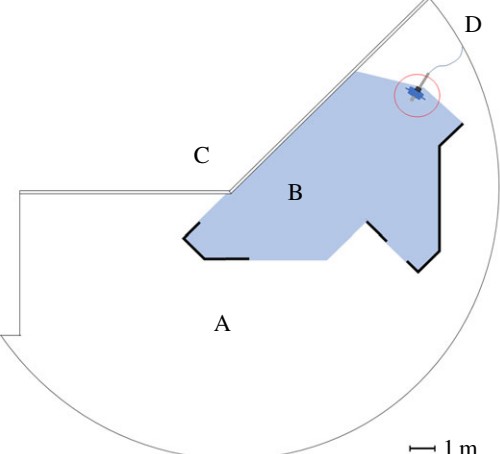

**Figure 1.** Drawing of the penguin enclosure in Odense Zoo. Playback set-up is pointing towards the pool centre (encircled with red). A, deck area; B, pool; C, public viewing area; D, experimenter's position.

To understand if and how penguins respond to underwater sound, we played back underwater noise bursts to diving gentoo penguins in a large pool. The penguins showed clear responses to the noise bursts, raising questions on how they use hearing while diving, as well as on how anthropogenic noise may affect them.

## 2. Material and methods

Underwater playbacks were conducted in a pool in the penguin enclosure at Odense Zoo, Denmark. There were three species of penguins living in the enclosure: 21 king penguins (*Aptenodytes patagonicus*), 6 northern rockhoppers (*Eudyptes moseleyi*) and 11 gentoo penguins (*Pygoscelis papua*). During playback sessions, only gentoo penguins were in the pool, whereas the other species remained on land. A maximum of three individuals were in the pool during each trial to minimize potential disturbances and noise from conspecifics. The birds were 6–24 years old at the time of the experiments. Two years earlier, some of them had been exposed to underwater sound of similar intensity and type as in the present trials, during a shorter pilot study, but otherwise they had no previous experience with experimental underwater sound exposure. None of them had previously been exposed to ototoxic medicine.

The pool was an irregularly shaped 175 m$^3$ pool (surface area 50 m$^2$, water depth 3.5 m, salinity 28‰, water temperature 6°C). The irregular shape of the pool (figure 1) ensured that the sound field was not severely affected by reflective planar surfaces, which may create long 'church-bell-like' reverberations in more regularly shaped pools.

Underwater playbacks were made on seven adult gentoo penguins from 13 July to 18 October 2018. Signals were noise bursts of 500 ms duration and filtered with an eighth-order Butterworth filter between 0.2 and 6 kHz, designed in Adobe Audition. Each signal had a 100 ms ramp-on and 100 ms ramp-off. Five stimulus levels were tested; 100, 105, 110, 115 and 120 dB re 1 µPa RMS; all source levels were measured 1 m from the speaker. The received level was then calculated for the position of the bird, resulting in a maximum received level variation of ±3 dB; see details below). Each sound level was played back eight times. In addition, we included eight control trials on three individual birds with no sound, randomly inserted between sound playback trials. Control trials followed exactly the same experimental and analysis protocol as stimulus trials, playing back a sound file that contained no signal. The signals were played back from a WAV file (sampling rate 48 kHz, 16 bits) in randomized order through a University Sound UW-30 underwater loudspeaker attached to a vertical PVC pipe at a depth of 1 m and connected to a laptop computer with Adobe Audition. An underwater video camera (Divers pro fish-eye 10–021, LH-video, Kolding, Denmark) was mounted inside the PVC pipe at 3 m depth close to the bottom and right below the loudspeaker, facing upwards to the surface and the speaker. The camera was connected via an Elgato video capture USB device to another laptop computer which recorded both the video signal and the emitted acoustic signals. A GoPro Hero 5 camera was mounted in an underwater housing right above the speaker and was used to determine which individual penguin was exposed in each trial.

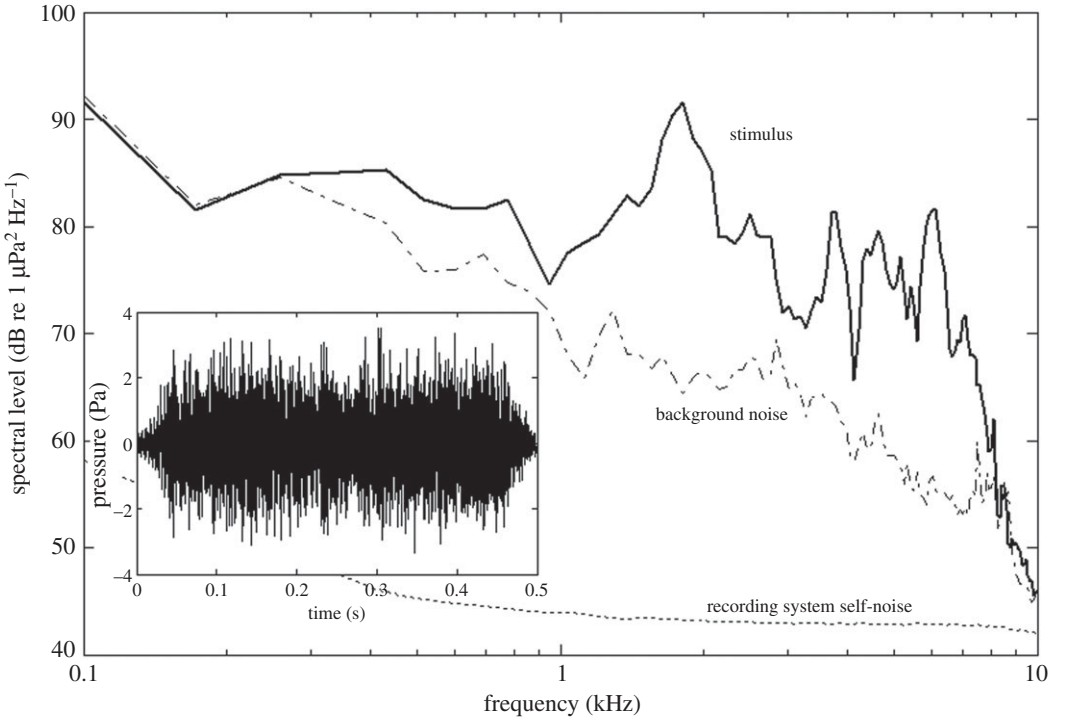

**Figure 2.** Oscillogram and power spectral density plot (solid line) of received 120 dB re 1 µPa signal and ambient noise (stippled line), measured 1 m in front of the loudspeaker. The dotted line is the recording system self-noise.

To measure stimulus levels and characterize spatial variability in the received stimuli, a SoundTrap (ST300HF, Ocean Instruments, Inc., sampling rate 48 kHz, 16 bits) acoustic data logger was held at different ranges, angles and depths in 0.5 m steps from 0.5 to 2 m range, and 1 m to the left and right and up and down relative to the acoustic axis from the speaker. The measurements encompassed the volume of water in front of the speaker where the bird was located at the time of playback. The stimulus was measured three times during the study: before, during and after the completion of the experiments. Ambient noise recordings were made throughout the playback period by holding the SoundTrap at 1 m depth and at 1 m in front of the loudspeaker for 1 min. The logger's saturation level in the frequency range of 0.1–20 kHz was determined with relative calibration in air (100–500 Hz) and under water (0.5–20 kHz) to be 171 dB re 1 µPa RMS within ±3 dB. Recorded playback signals and ambient noise recordings were analysed with custom-made programs in Matlab (v. 2017). The RMS level of recorded playback signals was calculated from equations in [31], and the noise spectrum was calculated with Welch spectral averaging (Hanning window, FFT size 256, 50% overlap). To estimate the logger's electric self-noise, the logger was left recording in a soundproof chamber for 1 min, after which the file was analysed using the same spectral averaging parameters given above for the tank noise recordings.

The penguin's range from the transducer was estimated for each playback trial by estimating the vertical and horizontal range and adding them up by Pythagoras' theorem. The range was estimated from video footage from the underwater video camera facing upwards to the surface. The horizontal range was estimated by comparing the measured distance from the bird to the loudspeaker with the known length of the penguin (62–72 (min/max) cm from feet to the tip of the beak). The vertical range was estimated by comparing the observed length of the penguin at 1 m range with the length measured during playback. The range from the penguin to the loudspeaker at playback was 0.4–1.5 m. The calibration of the sound field showed that the received level differed no more than 3 dB within the possible locations of the penguin at playback. Therefore, the measured source levels were used for analysis. The ambient noise levels (figure 2) were lower than all the stimuli's received levels in the frequency band where they overlapped, even though for the 100 dB re 1 µPa trials, the signal to noise ratio was poor.

The experimenter was positioned behind a wall out of sight of the penguins. He manually controlled the transmission of sound and the recording of the video while observing the swimming penguins on a computer screen, from the underwater camera. A maximum of three penguins were allowed in the pool simultaneously during testing. When a penguin swam by within the camera's visual field and within a

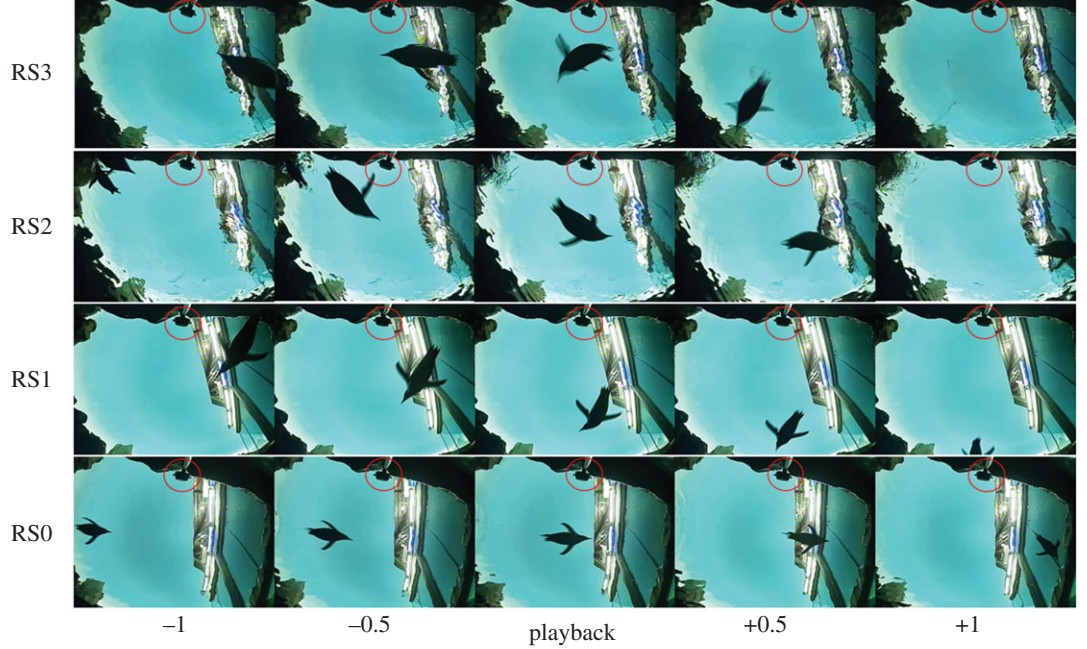

**Figure 3.** Video frame sequences of a trial with no response (RS0) at 100 dB re 1 µPa, a minor behavioural response (RS1) at 110 dB re 1 µPa, a clear behavioural response (RS2) at 120 dB re 1 µPa and a startle response (RS3) at 120 dB re 1 µPa. The sequences show the penguin 1 s before until 1 s after playback. The transducer (encircled with red) is at the top of each frame directed horizontally at 1 m depth.

**Table 1.** Definitions of response scores 0–3.

| score | name | definition |
| --- | --- | --- |
| RS0 | no response | no head movement or change in direction and speed |
| RS1 | minor behavioural response | 45–90° change of head movement |
| RS2 | clear behavioural response | 45–90° change of swim direction and change of speed |
| RS3 | startle response | >90° change of swim direction and change of speed |

range of 0.4–1.5 m to the loudspeaker, the experimenter would play back the signal. Only one penguin was within the camera's visual field during each trial, and any other penguins in the pool were at longer (at least 4 m) ranges from the loudspeaker. The sound level chosen for each trial was given by a random table. Each trial was followed by a pause (average 16 min, range 6–55 min) before a new trial was initiated. A maximum of five trials was made each day to avoid habituation to the stimuli. In total, 12 playback sequences were eventually omitted from the analysis due to the target penguin being too close or too far away from the speaker. An additional five sequences were omitted due to the penguins swimming behaviour (such as changing direction immediately before playback).

Videos were analysed using VideoLAN VLC Media Player. The signals transmitted during testing were connected to the video camera to synchronize video and sound signals. The initial analysis was made by the experimenter, omitting all sequences where the bird at the time of exposure was closer than 40 cm and further than 1.5 m from the loudspeaker. The experimenter also noted which individual bird was involved in each playback trial. Behavioural responses were graded by the experimenter and two experienced observers using Response Scores (RS) from 0 to 3 (defined in table 1, with examples given in figure 3). Responses were graded in a blind manner by both experimenter and the two additional observers, without them knowing either the exposed source level of each trial or which ones were control trials. The three observers did not significantly disagree from each other in their scores (Kruskal–Wallis test, d.f. = 7, $p > 0.05$); therefore, the average score from the observers was used in the analysis presented here.

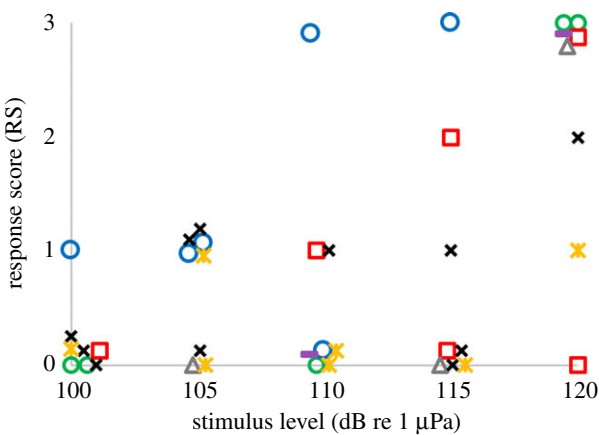

**Figure 4.** Response to playback by seven gentoo penguins (each individual indicated with different colours and shapes). Overlapping data points have been jittered for clarity.

**Table 2.** Number of playback trials for each of the seven gentoo penguins.

| sound level (dB re 1 µPa) | individual | | | | | | |
|---|---|---|---|---|---|---|---|
| | A | B | C | D | E | F | G |
| 100 | 0 | 1 | 3 | 1 | 0 | 2 | 1 |
| 105 | 0 | 2 | 3 | 2 | 1 | 0 | 0 |
| 110 | 1 | 2 | 1 | 2 | 0 | 1 | 1 |
| 115 | 0 | 1 | 3 | 1 | 1 | 0 | 2 |
| 120 | 1 | 1 | 1 | 0 | 1 | 2 | 2 |

After analysing the reactions of individual birds, all response data was pooled into a response index (where RS0 was weighted with a 0, RS1 with a 1, and so on) for every sound intensity. A linear regression was made between response index and received level, and whether the regression line slope was larger than zero (indicating a relationship between sound intensity and response) was tested with ANOVA. In addition, a binominal test was made by comparing the response of a pooled 'no response' (RS0 and 1) with a pooled 'response' (RS2 and 3). The R script used for this is available from [32].

## 3. Results

Results of playbacks on seven adult gentoo penguins are presented in figure 4. It was not possible to control for which individual was exposed in each trial, albeit each individual bird could be determined afterwards for each playback trial using the GoPro camera. This resulted in not all individuals being exposed to the same number of playbacks at all sound levels (table 2).

The response to playback was easily fitted into the four RS categories, illustrated in figure 3. There was no change in the response of the birds between subsequent playback trials, which could have indicated habituation to the sound emissions.

There was a clear graded response to the playback. This was obvious both when averaging the response scores of all birds, and when analysing the scores for individual birds. At exposure levels of 100 and 105 dB re 1 µPa, only responses RS1 and RS2 were observed. At 110 and 115 dB re 1 µPa, there was one RS3, whereas at 120 dB re 1 µPa, there was RS3 in more than 60% of the trials, and RS2 and RS3 responses in more than 75% of the trials (figure 5). All eight control trials were classified as RS0, with no response from the penguins.

The linear regression of the response and the intensity was significantly positive (ANOVA, d.f. = 4, $p < 0.05$), and there was a significant effect from the sound level on the pooled RS0–1 responses tested against RS2–3 responses (binominal test, $p < 0.05$).

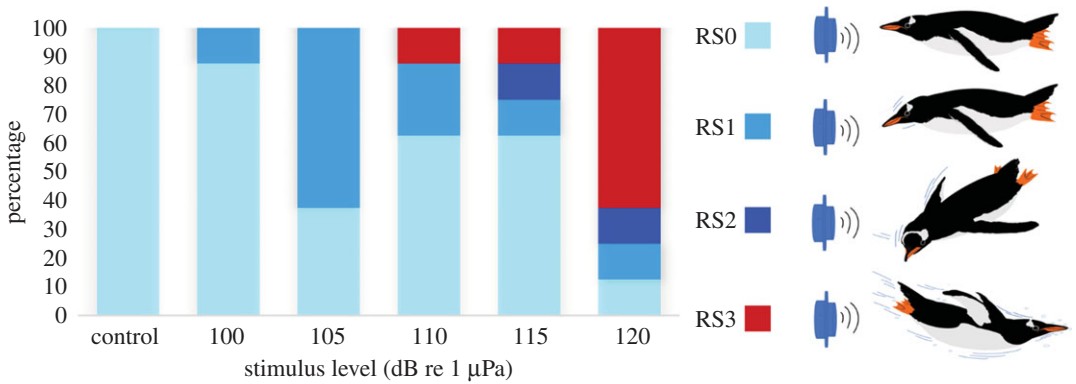

**Figure 5.** Stacked bar chart showing response scores (RS0–3) for different stimulus levels and controls. Each colour represents different response scores.

## 4. Discussion

All seven gentoo penguins reacted to the underwater noise pulses, and they did so in a graded manner. Received levels below 115 dB re 1 µPa RMS did not show any strong responses, whereas the highest playback intensities of 115 and 120 dB re 1 µPa elicited many strong responses. Combining the graded response to increased sound levels with the lack of responses for the control trials, this clearly showed that the penguins reacted to underwater sounds (figures 4 and 5). The stimulus contained frequencies between 0.2 and 6 kHz. This encompasses the known hearing range of penguins in air [15] and makes it likely that the frequency range of underwater hearing is similar. Judging by the gentoo penguins' fast swimming and curious behaviour, the extension of the pause between playback trials, as well as the restriction on the maximum number of playbacks per day, seemed to have been adequate to avoid any habituation to the signals. Indeed, there was no sign in the data of habituation, and the penguins usually reverted to their playful behaviour almost immediately after playback. In addition, noise from bubbles released from the plumage while diving can potentially mask the bird's ability to hear the stimulus, but no bubbles were observed at the time of any of the playbacks.

If we assume penguins have similar underwater hearing thresholds as the cormorant (about 70–75 dB re 1 µPa) [21], we expect the reaction threshold found in our study of some 115 dB re 1 µPa to be 45–50 dB above the audiogram. This is surprising low. Comparing with humans, a sound intensity 50 dB above the hearing threshold corresponds to speaking with a very soft voice and, therefore, not usually causing any strong behavioural reactions. Also, behavioural reactions of marine mammals are usually observed at received levels way beyond 100 dB relative to the hearing threshold [33], with the exception of the harbour porpoise (*Phocoena phocoena*) sometimes showing reaction thresholds less than 40 dB relative the hearing threshold [29]. The fact that penguins react strongly towards these rather weak sounds may either indicate that they strongly disliked the noise burst or that their underwater hearing threshold is even lower than the cormorant's. These explanations assume that penguin hearing is adapted to the underwater environment.

Another surprise was that the 20 dB increment in received level was sufficient to change the reaction of the birds from no observable response to a strong aversive response. This may be explained by the lower received levels being very close to the ambient noise levels in the tank (figure 2), resulting in a poor received signal to noise ratio for the bird. The ambient noise received by a moving bird may be even larger than what could be measured here due to flow noise around the bird. However, the tested penguins are used to the sound in the pool, and, therefore, the low increase in noise level that was sufficient to elicit a response is noteworthy. It may again hint at underwater sound stimuli being more important to penguins than previously assumed.

All observed startle responses (RS3) were directed away from the loudspeaker (see figure 3 for response examples). There are several possible explanations for this. Penguins may have an acute sense of directional hearing under water. It is also possible that, while swimming through the sound field, the penguin could detect the increasing sound level as it approaches the loudspeaker, and thereby responded by swimming towards lower sound levels. Finally, the penguin may have learned to associate the loudspeaker with sound emissions, so that its directional response away from the speaker is triggered by visual cues.

If penguins have directional hearing abilities under water, they must rely on some interesting adaptations in their hearing system. In air, directional hearing is achieved by a combination of temporal and intensity cues [19]. Both these cues are challenged under water. The higher speed of sound makes differences between the sound's time of arrival at the two ears much smaller. Also, the wavelengths at a given frequency are longer under water, which will result in a poorer shielding by the skull of the sound arriving at one ear and the other. Marine mammals seem to have solved these issues by having very acute time measurement abilities as well as (at least in the case of whales) an inner ear shielded by air sacs [34]. In birds, in-air directional hearing is accomplished by a pressure gradient coupling of the two ears [19]. It is unlikely that the same mechanism would work under water, and it is, therefore, currently unclear how penguins may be able to tease out the direction to an underwater sound source. Alternatively, if penguins rely on the intensity gradient or visual cues to perform the directional response to the speaker rather than on directional hearing abilities, this still indicates that these animals can detect and process underwater sounds.

The strong startle responses also indicate that penguins might react to novel sounds under water by fear rather than by curiosity. Penguins are known to be preyed upon by multiple predators, such as orcas and seals; some of the sounds these predators produce are of similar duration and bandwidth as the ones used in the playback trials [35]. There are also many abiotic events generating broadband noise in their underwater environment, such as ice floes moving along each other, falling and tipping blocks of ice, and seismic events. Such events are all potentially dangerous and may therefore be many good reasons for penguins to avoid broadband underwater noise sources. Penguins have also been shown to forage in groups [36] and may use acoustic cues either to detect prey or coordinate prey capture in a group.

The fact that penguins react strongly to broadband underwater sounds indicates that acoustic cues may be of great importance to penguins and other diving birds. It also highlights the concern that marine birds, just as whales and seals, may be vulnerable to man-made underwater noise sources. For example, the observations of penguins being affected by seismic exploration [37] could readily be explained by aversive reactions to the noise from such activities.

Ethics. This study was performed in accordance with relevant guidelines and regulations for animal experimentation at University of Southern Denmark (https://www.sdu.dk/-/media/sidste_chance/files/om_sdu/institutter/biolab/ethical+guidelines+for+the+use+of+animals+in+teaching+and+research+at+the+university+of+southern+den.pdf) and the Danish Animal Ethics Committee (https://www.foedevarestyrelsen.dk/english/Animal/AnimalWelfare/The-Animal-Experiments-Inspectorate/Pages/default.aspx). The study was not required to complete an ethical assessment prior to conducting the experiments.

Data accessibility. Data and R script [32] used for the analysis performed in this study are available on the Dryad Digital Repository: https://doi.org/10.5061/dryad.cc2fqz62v [32].

Authors' contributions. K.S. collected all data, participated in data analysis, participated in the design of the study and drafted the manuscript. M.W. supervised the project, participated in the design of the study and helped draft the manuscript. K.A.H. and M.D. participated in data analysis and critically revised the manuscript. C.N. carried out a similar unpublished pilot study and critically revised the manuscript. All authors gave final approval for publication and agree to be held accountable for the work performed therein.

Competing interests. We have no competing interests.

Funding. Parts of the study were funded through the project 'Hearing in penguins' by the German Environmental Agency through the Federal Ministry for the Environment Nature Conservation and Nuclear Safety (grant no. FKZ3777182440).

Acknowledgements. We thank Simeon Smeele for help with the statistical analysis.

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
