## [Reviewer comments · Royal Society Open Science]

Review History

RSOS-191988.R0 (Original submission)

Review form: Reviewer 1

Is the manuscript scientifically sound in its present form?

Yes

Are the interpretations and conclusions justified by the results?

Yes

Is the language acceptable?

Yes

Do you have any ethical concerns with this paper?

No

Have you any concerns about statistical analyses in this paper?

No

Recommendation?

Accept with minor revision (please list in comments)

Comments to the Author(s)

This revised manuscript has addressed most of my concerns with the original submission. Many details have been added for clarity, and the updated (and new) tables and figures are very helpful. Some additional comments and suggestions follow, which should improve the manuscript further before publication.

Abstract, Line 18. Should read "Underwater hearing in birds has been studied..."

Introduction, Line 34. Suggest delete "extreme" and change "such as" to "which include."

Line 36. "Year around" should be "year-round."

Line 39. I think the next few paragraphs could be streamlined and made more linear to tell a cohesive story. It's confusing that this paragraph starts with a statement that sensory systems are adjusted to function in both media. Then the next few paragraphs seem to go back and forth about the challenges of a secondarily aquatic existence and whether or not marine mammals/diving birds MIGHT be adapted to deal with this, and the fact that they are.

Line 41. This statement is not supported by the rest of the paragraph, which suggests that the senses are adapted (to varying degrees) to function in both media. Could say "these senses function differently due to different properties..."

Line 48. This statement is only true if modifications aren't made to enable sound detection in water. I suggest focusing on the differences between air/water and the challenges this presents, and then how the sensory systems have adapted to function in both. The statements about senses being "compromised" or that they "do not function well" are misleading and contradict the main assertions of the Introduction.

Line 55. The relative impedance of the surrounding medium and the head is another important factor here.

Line 69. Should read "The acute hearing abilities of marine mammals make them vulnerable..."

Line 71-72. Suggest "For louder sound sources (e.g., pile-driving noise), marine mammals can be displaced by tens of kilometers [14]."

Line 77-79. Suggest "Sea otters (*Enhydra lutris*) have relatively low hearing thresholds..." since sea otters don't hear as well as pinnipeds or cetaceans. In general, I suggest using more specific terminology here and elsewhere. "Low thresholds," "hear well under water," and similar phrases are ambiguous.

Line 85. Should read airborne instead of air-born. Also suggest "They use airborne sounds for behaviors such as mating..."

Line 93. We don't get around to underwater hearing in marine birds (the focus of the paper) until very far along in the introduction. The other background information is helpful, but could be made more concise.

Line 96. Should read "...penguins, which are highly specialized..."

Line 102. Should read "but also for general orientation..."

Line 107. Suggest "...noise bursts, raising questions on how..."

Materials and methods, Line 113. Scientific names?

Line 123. Period is missing at the end of the sentence.

Line 130. Suggest "source levels" instead of "sound levels," since received levels were also measured elsewhere.

Line 145. Suggest "To measure stimulus levels and characterize spatial variability in the received stimuli, a SoundTrap..." How often/how many times was this calibration completed? Please provide additional details.

Line 147. It might be useful to point out here how this range of distances compares to the acceptable position of the bird during playbacks.

Line 158. Mention here that the range was estimated from video footage. Which camera was used?

Line 164-165. It would be helpful to add an explicit statement here to clarify that source levels were used. For example, after the sentence "The calibration of the sound field showed that the received level differed no more than 3 dB within the possible locations of the penguin at playback. Therefore, the measured source levels were used for analysis."

Line 175. Can any values be provided for this? what was the minimum distance between the focal bird and another individual?

Line 183. How many sequences were omitted?

Line 187. Did this include the experimenter? In other words, was the experimenter blind to the trial conditions before scoring the trial? If not, this must be specified.

Results, Line 206. Should be playback.

Line 210. There was no indication of the response of the birds changing between subsequent trials at the same level, or at different levels? I'm not sure that this statement is appropriate without more data or stats provided.

Discussion, Line 228. It would be helpful to remind the reader of the bandwidth of these stimuli here, and to discuss in relation to hearing range.

Line 234. The comparative example with humans is helpful, but what about other marine mammals? That information would be more relevant.

Line 249. This point is not clear based on the data presented.

Line 251. Should read penguins'

Line 254. Was a return to baseline behavior quantified in any way?

Line 257. Consider moving this paragraph up, to follow the one about the reaction threshold (starting with "If we assume penguins have similar underwater hearing thresholds as the cormorant...")

Line 262. Noise released from plumage during dives can be significant. Were bubbles being released continuously during all playbacks? Was this quantified? If not, it's still good to mention this here and the authors do a good job acknowledging these caveats.

Line 267. Suggest delete “aversive” here in relation to the startle response (rather than the stimuli).

Line 268. Suggest "There are several possible explanations for this. Penguins may have an acute sense of directional hearing under water. It is also possible that, while swimming through the sound field, the penguin could detect the increasing sound level as it approached the loudspeaker and therefore responded by...."

Line 276-280. These statements are repeated from the Introduction. suggest remove in one of these sections. If left in the introduction, this paragraph could be made much more concise here.

Line 287. Suggest combining these sentences: "Alternatively, if penguins rely on the intensity gradient or visual cues to perform the directional response to the speaker rather than relying on directional hearing abilities, this still indicates that these animals can detect and process underwater sounds." Could potentially combine with prior paragraph as well.

Line 292. Suggest strong instead of “aversive” here.

Line 295. I think this is a bit misleading, since orcas and seals can produce sounds of similar duration and bandwidth as these, but they also produce a wide variety of sounds. Suggest something like “Penguins are known to be preyed upon orcas and seals; some of the sounds these predators produce are of similar duration and bandwidth as the ones used in the playback trials”

Figure 1. What is the outer, white area in this figure – the pool deck? Where is the viewing area, where the public could be located? Suggest adding some labels to this figure.

Figure 2. What is the third line? Is this the self-noise of the instrumentation? All three lines must be described clearly. Suggest "Oscillogram and power spectral density plots of received 120 dB re 1 μ Pa signal (solid line), ambient noise (dashed line), and self-noise of the instrument (dotted line) measured..."Also need to fix the superscript in the y-axis units.

Figure 3 is very useful! It’s also helpful to present the pooled data (Figure 5) as well as the data by individual (Figure 4).

Review form: Reviewer 2

Is the manuscript scientifically sound in its present form?

Yes

Are the interpretations and conclusions justified by the results?

No

Is the language acceptable?

Yes

Do you have any ethical concerns with this paper?

No

Have you any concerns about statistical analyses in this paper?

No

Recommendation?

Accept with minor revision (please list in comments)

Comments to the Author(s)

This paper has been revised from its original form based on the suggestions of four Reviewers (myself included as one of these). In the Introduction, Methods, and Results, I think that the changes sufficiently address the previous concerns of the reviewers. I still have some problems with the Discussion. Perhaps these are stylistic differences, and I understand that the authors may want to put their results into a broader context or to inspire further research. However, as written, some of their conclusions do not have sufficient support. These are detailed below, along with some minor comments on the earlier parts of the paper.

Specific Comments

Line 52 Sound localization in the horizontal plane is based on these processes of time and intensity differences, but in the vertical plane it is more a function of filtering by the head and external ears. Suggest specifying.

Line 61 These modifications are evolutionary adaptations, not learned. Suggest clarifying.

Line 102 Do krill make sound? I couldn't find anything regarding this so a reference would be good here.

Line 119 Suggest changing to "experimental sound exposure" as the penguins have experienced underwater sound before.

Line 189 I'm not clear on the last sentence of this paragraph regarding controls, specifically the part about "one penguin". Was only one penguin used for control?

Line 233 I really think that this is an overreach of the data. Yes, humans would likely not react to speech at a received level of 50 dB for an expected source at some distance from the receiver. An unexpected source within 1 m of the head - especially noise which is often intrinsically perceived as more aversive than more tonal stimuli - may very well elicit a response. At low frequencies (due to loudness compression) 50 dB above threshold may be perceived to be as loud as something perhaps 80 dB above threshold in the range of best hearing sensitivity. Could these factors be contributing here? So, while a quick glance at the levels from the study may make the observations initially surprising, I think a more thorough discussion of them would at least cast some doubt on whether it is that much of a surprise.

Line 243 The statement that the similarity of the birds' reactions is a surprise seems to undercut the study. For a well-designed experiment, wouldn't you hope that there is some generalizability of the observed behavior among subjects? Very variable reactions would be much more surprising to me and would perhaps suggest that something is wrong with the methods, or that the birds have vastly different histories. Also, the last sentence of this paragraph is kind of a non sequitur, and it is unclear on what assumption of reaction the statement is based. Consistent reactions should not necessarily mean that they are "more prone to react", just that they are consistent in reacting.

Line 292 The statement that "penguins probably react to novel sounds under water by fear rather than by curiosity" can't be backed up by the current data, which are very specific in scope. This could be reworded in a way to highlight the topic of neophobic reactions to sound in penguins, but "probably" seems too much of a reach here.

Decision letter (RSOS-191988.R0)

14-Jan-2020

Dear Mr Sørensen

On behalf of the Editors, I am pleased to inform you that your Manuscript RSOS-191988 entitled "Gentoo penguins (*Pygoscelis papua*) react to underwater sounds" has been accepted for publication in Royal Society Open Science subject to minor revision in accordance with the referee suggestions. Please find the referees' comments at the end of this email.

The reviewers and handling editors have recommended publication, but also suggest some minor revisions to your manuscript. Therefore, I invite you to respond to the comments and revise your manuscript.

- Ethics statement

- Data accessibility

<http://datadryad.org/submit?journalID=RSOS&manu=RSOS-191988>

- Competing interests

- Authors' contributions

AB carried out the molecular lab work, participated in data analysis, carried out sequence alignments, participated in the design of the study and drafted the manuscript; CD carried out the statistical analyses; EF collected field data; GH conceived of the study, designed the study,

coordinated the study and helped draft the manuscript. All authors gave final approval for publication.

- Acknowledgements

- Funding statement

Because the schedule for publication is very tight, it is a condition of publication that you submit the revised version of your manuscript before 23-Jan-2020. Please note that the revision deadline will expire at 00.00am on this date. If you do not think you will be able to meet this date please let me know immediately.

If your manuscript is newly submitted and subsequently accepted for publication, you will be asked to pay the article processing charge, unless you request a waiver and this is approved by Royal Society Publishing. You can find out more about the charges at <https://royalsocietypublishing.org/rsos/charges>. Should you have any queries, please contact openscience@royalsociety.org.

on behalf of Dr Denise Greig (Associate Editor) and Kevin Padian (Subject Editor)
openscience@royalsociety.org

Associate Editor Comments to Author (Dr Denise Greig):

Associate Editor: 1

Comments to the Author:

Two of the original reviewers agreed that your revision addressed most of their comments, however they still had some concerns over the presentation and interpretation of the data which they have clearly detailed. Their comments and suggestions cover all the questions I had, and I look forward to reading the next version.

Reviewer comments to Author:

Reviewer: 1

Comments to the Author(s)

This revised manuscript has addressed most of my concerns with the original submission. Many details have been added for clarity, and the updated (and new) tables and figures are very helpful. Some additional comments and suggestions follow, which should improve the manuscript further before publication.

Abstract, Line 18. Should read "Underwater hearing in birds has been studied..."

Introduction, Line 34. Suggest delete "extreme" and change "such as" to "which include."

Line 36. "Year around" should be "year-round."

Line 39. I think the next few paragraphs could be streamlined and made more linear to tell a cohesive story. It's confusing that this paragraph starts with a statement that sensory systems are adjusted to function in both media. Then the next few paragraphs seem to go back and forth about the challenges of a secondarily aquatic existence and whether or not marine mammals/diving birds MIGHT be adapted to deal with this, and the fact that they are.

Line 41. This statement is not supported by the rest of the paragraph, which suggests that the senses are adapted (to varying degrees) to function in both media. Could say "these senses function differently due to different properties..."

Line 48. This statement is only true if modifications aren't made to enable sound detection in water. I suggest focusing on the differences between air/water and the challenges this presents, and then how the sensory systems have adapted to function in both. The statements about senses being "compromised" or that they "do not function well" are misleading and contradict the main assertions of the Introduction.

Line 55. The relative impedance of the surrounding medium and the head is another important factor here.

Line 69. Should read "The acute hearing abilities of marine mammals make them vulnerable..."

Line 71-72. Suggest "For louder sound sources (e.g., pile-driving noise), marine mammals can be displaced by tens of kilometers [14]."

Line 77-79. Suggest "Sea otters (*Enhydra lutris*) have relatively low hearing thresholds..." since sea otters don't hear as well as pinnipeds or cetaceans. In general, I suggest using more specific terminology here and elsewhere. "Low thresholds," "hear well under water," and similar phrases are ambiguous.

Line 85. Should read airborne instead of air-born. Also suggest "They use airborne sounds for behaviors such as mating..."

Line 93. We don't get around to underwater hearing in marine birds (the focus of the paper) until very far along in the introduction. The other background information is helpful, but could be made more concise.

Line 96. Should read "...penguins, which are highly specialized..."

Line 102. Should read "but also for general orientation..."

Line 107. Suggest "...noise bursts, raising questions on how..."

Materials and methods, Line 113. Scientific names?

Line 123. Period is missing at the end of the sentence.

Line 130. Suggest "source levels" instead of "sound levels," since received levels were also measured elsewhere.

Line 145. Suggest "To measure stimulus levels and characterize spatial variability in the received stimuli, a SoundTrap..." How often/how many times was this calibration completed? Please provide additional details.

Line 147. It might be useful to point out here how this range of distances compares to the acceptable position of the bird during playbacks.

Line 158. Mention here that the range was estimated from video footage. Which camera was used?

Line 164-165. It would be helpful to add an explicit statement here to clarify that source levels were used. For example, after the sentence "The calibration of the sound field showed that the received level differed no more than 3 dB within the possible locations of the penguin at playback. Therefore, the measured source levels were used for analysis."

Line 175. Can any values be provided for this? what was the minimum distance between the focal bird and another individual?

Line 183. How many sequences were omitted?

Line 187. Did this include the experimenter? In other words, was the experimenter blind to the trial conditions before scoring the trial? If not, this must be specified.

Results, Line 206. Should be playback.

Line 210. There was no indication of the response of the birds changing between subsequent trials at the same level, or at different levels? I'm not sure that this statement is appropriate without more data or stats provided.

Discussion, Line 228. It would be helpful to remind the reader of the bandwidth of these stimuli here, and to discuss in relation to hearing range.

Line 234. The comparative example with humans is helpful, but what about other marine mammals? That information would be more relevant.

Line 249. This point is not clear based on the data presented.

Line 251. Should read penguins'

Line 254. Was a return to baseline behavior quantified in any way?

Line 257. Consider moving this paragraph up, to follow the one about the reaction threshold (starting with "If we assume penguins have similar underwater hearing thresholds as the cormorant...")

Line 262. Noise released from plumage during dives can be significant. Were bubbles being released continuously during all playbacks? Was this quantified? If not, it's still good to mention this here and the authors do a good job acknowledging these caveats.

Line 267. Suggest delete "aversive" here in relation to the startle response (rather than the stimuli).

Line 268. Suggest "There are several possible explanations for this. Penguins may have an acute sense of directional hearing under water. It is also possible that, while swimming through the sound field, the penguin could detect the increasing sound level as it approached the loudspeaker and therefore responded by...."

Line 276-280. These statements are repeated from the Introduction. suggest remove in one of these sections. If left in the introduction, this paragraph could be made much more concise here.

Line 287. Suggest combining these sentences: "Alternatively, if penguins rely on the intensity gradient or visual cues to perform the directional response to the speaker rather than relying on

directional hearing abilities, this still indicates that these animals can detect and process underwater sounds." Could potentially combine with prior paragraph as well.

Line 292. Suggest strong instead of "aversive" here.

Line 295. I think this is a bit misleading, since orcas and seals can produce sounds of similar duration and bandwidth as these, but they also produce a wide variety of sounds. Suggest something like "Penguins are known to be preyed upon orcas and seals; some of the sounds these predators produce are of similar duration and bandwidth as the ones used in the playback trials"

Figure 1. What is the outer, white area in this figure - the pool deck? Where is the viewing area, where the public could be located? Suggest adding some labels to this figure.

Figure 2. What is the third line? Is this the self-noise of the instrumentation? All three lines must be described clearly. Suggest "Oscillogram and power spectral density plots of received 120 dB re 1 μ Pa signal (solid line), ambient noise (dashed line), and self-noise of the instrument (dotted line) measured..." Also need to fix the superscript in the y-axis units.

Figure 3 is very useful! It's also helpful to present the pooled data (Figure 5) as well as the data by individual (Figure 4).

Reviewer: 2

Comments to the Author(s)

This paper has been revised from its original form based on the suggestions of four Reviewers (myself included as one of these). In the Introduction, Methods, and Results, I think that the changes sufficiently address the previous concerns of the reviewers. I still have some problems with the Discussion. Perhaps these are stylistic differences, and I understand that the authors may want to put their results into a broader context or to inspire further research. However, as written, some of their conclusions do not have sufficient support. These are detailed below, along with some minor comments on the earlier parts of the paper.

Specific Comments

Line 52 Sound localization in the horizontal plane is based on these processes of time and intensity differences, but in the vertical plane it is more a function of filtering by the head and external ears. Suggest specifying.

Line 61 These modifications are evolutionary adaptations, not learned. Suggest clarifying.

Line 102 Do krill make sound? I couldn't find anything regarding this so a reference would be good here.

Line 119 Suggest changing to "experimental sound exposure" as the penguins have experienced underwater sound before.

Line 189 I'm not clear on the last sentence of this paragraph regarding controls, specifically the part about "one penguin". Was only one penguin used for control?

Line 233 I really think that this is an overreach of the data. Yes, humans would likely not react to speech at a received level of 50 dB for an expected source at some distance from the receiver. An unexpected source within 1 m of the head - especially noise which is often intrinsically perceived as more aversive than more tonal stimuli - may very well elicit a response. At low frequencies (due to loudness compression) 50 dB above threshold may be perceived to be as loud as something perhaps 80 dB above threshold in the range of best hearing sensitivity. Could these

factors be contributing here? So, while a quick glance at the levels from the study may make the observations initially surprising, I think a more thorough discussion of them would at least cast some doubt on whether it is that much of a surprise.

Line 243 The statement that the similarity of the birds' reactions is a surprise seems to undercut the study. For a well-designed experiment, wouldn't you hope that there is some generalizability of the observed behavior among subjects? Very variable reactions would be much more surprising to me and would perhaps suggest that something is wrong with the methods, or that the birds have vastly different histories. Also, the last sentence of this paragraph is kind of a non sequitur, and it is unclear on what assumption of reaction the statement is based. Consistent reactions should not necessarily mean that they are "more prone to react", just that they are consistent in reacting.

Line 292 The statement that "penguins probably react to novel sounds under water by fear rather than by curiosity" can't be backed up by the current data, which are very specific in scope. This could be reworded in a way to highlight the topic of neophobic reactions to sound in penguins, but "probably" seems too much of a reach here.

Author's Response to Decision Letter for (RSOS-191988.R0)

See Appendix A.

Decision letter (RSOS-191988.R1)

28-Jan-2020

Dear Mr Sørensen:

On behalf of the Editors, I am pleased to inform you that your Manuscript RSOS-191988.R1 entitled "Gentoo penguins (*Pygoscelis papua*) react to underwater sounds" has been accepted for publication in Royal Society Open Science subject to minor revision in accordance with the referee suggestions. Please find the referees' comments at the end of this email.

The Associate Editor has recommended publication, but also suggest some minor revisions to your manuscript. Therefore, I invite you to respond to the comments and revise your manuscript.

- Ethics statement

- Data accessibility

It is a condition of publication that all supporting data are made available either as supplementary information or preferably in a suitable permanent repository. The data accessibility section should state where the article's supporting data can be accessed. This section should also include details, where possible of where to access other relevant research materials

such as statistical tools, protocols, software etc can be accessed. If the data has been deposited in an external repository this section should list the database, accession number and link to the DOI for all data from the article that has been made publicly available. Data sets that have been deposited in an external repository and have a DOI should also be appropriately cited in the manuscript and included in the reference list.

If you wish to submit your supporting data or code to Dryad (<http://datadryad.org/>), or modify your current submission to dryad, please use the following link:
<http://datadryad.org/submit?journalID=RSOS&manu=RSOS-191988.R1>

- **Competing interests**

- **Authors' contributions**

- **Acknowledgements**

- **Funding statement**

Because the schedule for publication is very tight, it is a condition of publication that you submit the revised version of your manuscript before 06-Feb-2020. Please note that the revision deadline will expire at 00.00am on this date. If you do not think you will be able to meet this date please let me know immediately.

When submitting your revised manuscript, you will be able to respond to the comments made by the referees and upload a file "Response to Referees" in "Section 6 - File Upload". You can use this

to document any changes you make to the original manuscript. In order to expedite the processing of the revised manuscript, please be as specific as possible in your response to the referees.

on behalf of Dr Denise Greig (Associate Editor) and Kevin Padian (Subject Editor)
openscience@royalsociety.org

Associate Editor Comments to Author (Dr Denise Greig):

Thank you for the additional revisions and your response to the reviewers! The experiment and the discussion are much clearer now, and the results are very interesting. I am attaching a few minor edits in track changes, and thank you for your contribution to RSOS.

Author's Response to Decision Letter for (RSOS-191988.R1)

See Appendix B.

Decision letter (RSOS-191988.R2)

31-Jan-2020

Dear Mr Sørensen,

It is a pleasure to accept your manuscript entitled "Gentoo penguins (*Pygoscelis papua*) react to underwater sounds" in its current form for publication in Royal Society Open Science.

Kind regards,
Lianne Parkhouse
Editorial Coordinator
Royal Society Open Science
openscience@royalsociety.org

on behalf of Dr Denise Greig (Associate Editor) and Kevin Padian (Subject Editor)
openscience@royalsociety.org

Odense, 19th of January, 2020

Dear Editor,

We are thrilled about the possibility to have our work published in RSOS.

Please find attached a revised version of the manuscript *Gentoo penguins (Pygoscelis papua) react to underwater sounds*, taking the issues raised by the reviewers into account. Our comments on reviewers' issues and how we have dealt with them are found below.

Please do not hesitate to contact us if there are any further issues.

Sincerely,

Kenneth Sørensen and coauthors

Comments on reviewers' issues, Sørensen et al., submitted to RSOS**Reviewer 1**

This revised manuscript has addressed most of my concerns with the original submission. Many details have been added for clarity, and the updated (and new) tables and figures are very helpful. Some additional comments and suggestions follow, which should improve the manuscript further before publication.

Abstract, Line 18. Should read "Underwater hearing in birds has been studied..."

- *OK, 'marine' has been omitted.*

Introduction, Line 34. Suggest delete "extreme" and change "such as" to "which include."

- *OK, changed.*

Line 36. "Year around" should be "year-round."

- *OK, changed.*

Line 39. I think the next few paragraphs could be streamlined and made more linear to tell a cohesive story. It's confusing that this paragraph starts with a statement that sensory systems are adjusted to function in both media. Then the next few paragraphs seem to go back and forth about the challenges of a secondarily aquatic existence and whether or not marine mammals/diving birds MIGHT be adapted to deal with this, and the fact that they are.

- *We have changed the order and wording of these paragraphs to improve the flow:*
 - *First, we introduce issues with hearing underwater*
 - *Then explain about marine mammal adaptations*
 - *Third, we tell about what is known in other secondarily aquatically adapted species*
 - *Then we move to the marine birds:*
 - *Hearing in air*
 - *Hearing underwater*
 - *Possible effects of underwater noise (this paragraph has now been changed to have more of a bird-focus)*

Line 41. This statement is not supported by the rest of the paragraph, which suggests that the senses are adapted (to varying degrees) to function in both media. Could say "these senses function differently due to different properties..."

- *OK, changed.*

Line 48. This statement is only true if modifications aren't made to enable sound detection in water. I suggest focusing on the differences between air/water and the challenges this presents, and then how the sensory systems have adapted to function in both. The statements about senses being "compromised" or that they "do not function well" are misleading and contradict the main assertions of the Introduction.

- *True. This has been clarified and the new first sentence of this paragraph reads: 'Besides vision, the sense of hearing of an air-adapted animal is also challenged.'*

Line 55. The relative impedance of the surrounding medium and the head is another important factor here.

- *It is true that the relative impedance is important for sound detection, but we maintain that the two major issues for sound localization are the phase and amplitude differences between the two ears – so we have kept the sentence as-is.*

Line 69. Should read "The acute hearing abilities of marine mammals make them vulnerable..."

- *True; changed.*

Line 71-72. Suggest “For louder sound sources (e.g., pile-driving noise), marine mammals can be displaced by tens of kilometers [14].”

- *OK, changed.*

Line 77-79. Suggest “Sea otters (*Enhydra lutris*) have relatively low hearing thresholds...” since sea otters don’t hear as well as pinnipeds or cetaceans. In general, I suggest using more specific terminology here and elsewhere. “Low thresholds,” “hear well under water,” and similar phrases are ambiguous.

- *OK, changed for the sea otter, and ‘low’ and ‘hear well’ has been specified better for the other species. The entire paragraph now reads:
‘Besides whales and seals, little is known about hearing adaptations to the aquatic environment in secondarily adapted aquatic vertebrates. Sea otters (*Enhydra lutris*) have relatively low hearing thresholds in a broad frequency range in both media [16], and manatees (*Trichechus manatus*) hear comparably well under water [17]. Some aquatic turtles (e.g., *Trachemys scripta*) have tuned their hearing to function in both media, even though the sensitivity and frequency range is restricted compared to marine mammals [18]. Polar bears (*Ursus maritimus*) have hearing abilities comparable to other similar-sized mammals on land [19], but their hearing abilities under water have not been studied.’*

Line 85. Should read airborne instead of air-born. Also suggest “They use airborne sounds for behaviors such as mating...”

- *Thanks; corrected.*

Line 93. We don’t get around to underwater hearing in marine birds (the focus of the paper) until very far along in the introduction. The other background information is helpful, but could be made more concise.

- *OK. We have now flipped around the order of the paragraphs a bit (see comment to Line 39 above) and also included some more statements of the birds in some of the ‘mammal paragraphs’.*

Line 96. Should read “...penguins, which are highly specialized...”

- *OK, changed.*

Line 102. Should read “but also for general orientation...”

- *OK, changed.*

Line 107. Suggest “...noise bursts, raising questions on how...”

- *OK, changed.*

Materials and methods, Line 113. Scientific names?

- *Scientific names have been added.*

Line 123. Period is missing at the end of the sentence.

- *Thanks; period has been added.*

Line 130. Suggest “source levels” instead of “sound levels,” since received levels were also measured elsewhere.

- *OK, changed.*

Line 145. Suggest “To measure stimulus levels and characterize spatial variability in the received stimuli, a SoundTrap...” How often/how many times was this calibration completed? Please provide additional details.

- *We measured levels three times: before the start of the experiments, in the middle of them, and after the experiments were completed. This has been included in the new version of the manuscript.*

Line 147. It might be useful to point out here how this range of distances compares to the acceptable position of the bird during playbacks.

- *True, this has been clarified further down, where calibrations are explained in detail.*

Line 158. Mention here that the range was estimated from video footage. Which camera was used?

- *This was made using the underwater video camera facing upwards. This has been explicitly explained in the new version of the ms.*

Line 164-165. It would be helpful to add an explicit statement here to clarify that source levels were used. For example, after the sentence “The calibration of the sound field showed that the received level differed no more than 3 dB within the possible locations of the penguin at playback. Therefore, the measured source levels were used for analysis.”

- *Ok, this has been included.*

Line 175. Can any values be provided for this? what was the minimum distance between the focal bird and another individual?

- *The shortest distance to another bird was 4 m. This has been included in the text.*

Line 183. How many sequences were omitted?

- *12 sequences were omitted due to penguins being too close/too far away from the speaker. Additionally, 5 sequences were omitted due to penguins swimming behavior (changing direction during playback etc.) This has been included in the new version.*

Line 187. Did this include the experimenter? In other words, was the experimenter blind to the trial conditions before scoring the trial? If not, this must be specified.

- *The experimenter was blind to the trial conditions before scoring the trial, but during the actual trials, he was not, due to the way the experiments were administrated. He was however always out of vision from the penguins and could therefore not affect their behaviour in any way. This information has been added to the new version.*

Results, Line 206. Should be playback.

- *OK, changed.*

Line 210. There was no indication of the response of the birds changing between subsequent trials at the same level, or at different levels? I’m not sure that this statement is appropriate without more data or stats provided.

- *We would like to maintain this information. Even though the data set is too small to allow a full study of habituation, the indication that no change in the response was observed put confidence in the data being collected without any severe habituation happening during the experimental period.*

Discussion, Line 228. It would be helpful to remind the reader of the bandwidth of these stimuli here, and to discuss in relation to hearing range.

- *OK, this has been included.*

Line 234. The comparative example with humans is helpful, but what about other marine mammals? That information would be more relevant.

- OK, included in the new version:

*'Also, behavioural reactions on marine mammals are usually observed at received levels way beyond 100 dB relative the hearing threshold [Southall et al. 2007], the only exemption seemingly being the harbour porpoise (*Phocoena phocoena*) sometimes showing reaction thresholds less than 40 dB relative the hearing threshold (Tougaard et al. 2015).'*

Line 249. This point is not clear based on the data presented.

- OK; it has been omitted and because this is the key sentence of the entire paragraph, we have deleted this paragraph.

Line 251. Should read penguins'

- OK, changed, thanks.

Line 254. Was a return to baseline behavior quantified in any way?

- No, baseline was just characterized as calm and playful behavior but not quantified.

Line 257. Consider moving this paragraph up, to follow the one about the reaction threshold (starting with "If we assume penguins have similar underwater hearing thresholds as the cormorant...")

- Given that we erased one paragraph (see Line 249 comment), the only paragraph remaining between the 'If we assume...' and 'Another surprise...' is the paragraph:
'Judging by the gentoo penguins' fast swimming and curious behaviour, the extension of the pause between playback trials, as well as the restriction on the maximum number of playbacks per day, seemed to have been adequate to avoid any habituation to the signals. Indeed, there was no sign in the data on habituation to occur, and the penguins usually reverted to their playful behaviour almost immediately after playback.'
To make the discussion flow better (see also comments from Reviewer 2), we have moved this paragraph to the final part of the discussion's first paragraph.

Line 262. Noise released from plumage during dives can be significant. Were bubbles being released continuously during all playbacks? Was this quantified? If not, it's still good to mention this here and the authors do a good job acknowledging these caveats.

- We did not observe any release of bubbles during playbacks. We have included a sentence about this at the end of the first paragraph of the discussion. We have also omitted the sentence in the paragraph of reviewer's concern here dealing with bubble release, as it was not an issue in our study.

Line 267. Suggest delete "aversive" here in relation to the startle response (rather than the stimuli).

- OK, deleted.

Line 268. Suggest "There are several possible explanations for this. Penguins may have an acute sense of directional hearing under water. It is also possible that, while swimming through the sound field, the penguin could detect the increasing sound level as it approached the loudspeaker and therefore responded by...."

- OK, changed.

Line 276-280. These statements are repeated from the Introduction. suggest remove in one of these sections. If left in the introduction, this paragraph could be made much more concise here.

- *We think the only clear overlap between introduction and discussion is the sentence discussing sounds from prey, which has now been omitted from the introduction: 'This could be advantageous both to find prey, such as fish and krill, but also for general orientation under water, especially in situations where light conditions are limited.'*

Line 287. Suggest combining these sentences: "Alternatively, if penguins rely on the intensity gradient or visual cues to perform the directional response to the speaker rather than relying on directional hearing abilities, this still indicates that these animals can detect and process underwater sounds." Could potentially combine with prior paragraph as well.

- *OK, thanks, we have changed this.*

Line 292. Suggest strong instead of "aversive" here.

- *OK, we changed it to 'strong'.*

Line 295. I think this is a bit misleading, since orcas and seals can produce sounds of similar duration and bandwidth as these, but they also produce a wide variety of sounds. Suggest something like "Penguins are known to be preyed upon orcas and seals; some of the sounds these predators produce are of similar duration and bandwidth as the ones used in the playback trials"

- *OK, changed.*

Figure 1. What is the outer, white area in this figure – the pool deck? Where is the viewing area, where the public could be located? Suggest adding some labels to this figure.

- *We have added this information to the new figure.*

Figure 2. What is the third line? Is this the self-noise of the instrumentation? All three lines must be described clearly. Suggest "Oscillogram and power spectral density plots of received 120 dB re 1 μ Pa signal (solid line), ambient noise (dashed line), and self-noise of the instrument (dotted line) measured..." Also need to fix the superscript in the y-axis units.

- *OK, fixed all of it.*

Figure 3 is very useful! It's also helpful to present the pooled data (Figure 5) as well as the data by individual (Figure 4).

- *Thanks!*

Reviewer: 2

Comments to the Author(s)

This paper has been revised from its original form based on the suggestions of four Reviewers (myself included as one of these). In the Introduction, Methods, and Results, I think that the changes sufficiently address the previous concerns of the reviewers. I still have some problems with the Discussion. Perhaps these are stylistic differences, and I understand that the authors may want to put their results into a broader context or to inspire further research. However, as written, some of their conclusions do not have sufficient support. These are detailed below, along with some minor comments on the earlier parts of the paper.

Specific Comments

Line 52 Sound localization in the horizontal plane is based on these processes of time and intensity differences, but in the vertical plane it is more a function of filtering by the head and external ears. Suggest specifying.

- *OK – we have added –‘horizontal plane’ to make this absolutely clear*

Line 61 These modifications are evolutionary adaptations, not learned. Suggest clarifying.

- *‘learning’ has been omitted*

Line 102 Do krill make sound? I couldn’t find anything regarding this so a reference would be good here.

- *True that this has not been studied, but many crustaceans do produce sounds. However, sentence has been erased in response to rev. 1.*

Line 119 Suggest changing to “experimental sound exposure” as the penguins have experienced underwater sound before.

- *OK, ‘experimental’ has been added.*

Line 189 I’m not clear on the last sentence of this paragraph regarding controls, specifically the part about “one penguin”. Was only one penguin used for control?

- *Sorry for the confusion here. 3 penguins were used in control trials. This sentence has been deleted. A new sentence has been included to clarify the protocol of control trials: ‘Control trials followed exactly the same experimental and analysis protocol as stimulus trials, playing back a sound file that contained no signal.’ This sentence has been moved up to the paragraph describing the stimulus trials, where it more appropriately belongs.*

Line 233 I really think that this is an overreach of the data. Yes, humans would likely not react to speech at a received level of 50 dB for an expected source at some distance from the receiver. An unexpected source within 1 m of the head - especially noise which is often intrinsically perceived as more aversive than more tonal stimuli - may very well elicit a response. At low frequencies (due to loudness compression) 50 dB above threshold may be perceived to be as loud as something perhaps 80 dB above threshold in the range of best hearing sensitivity. Could these factors be contributing here? So, while a quick glance at the levels from the study may make the observations initially surprising, I think a more thorough discussion of them would at least cast some doubt on whether it is that much of a surprise.

- *We have added some examples from the marine mammal literature to elucidate this further, also as response to rev. 1. For marine mammals, there has been many playback studies, and except for one case (harbour porpoises) they show reactional thresholds well beyond 100 dB re hearing threshold. So, in spite of the reviewer’s concern regarding humans (which may or may not be valid in our view), we maintain that it is surprising with these consistently low and rather similar reaction thresholds from several individual penguins.*

Line 243 The statement that the similarity of the birds’ reactions is a surprise seems to undercut the study. For a well-designed experiment, wouldn’t you hope that there is some generalizability of the observed behavior among subjects? Very variable reactions would be much more surprising to me and would perhaps suggest that something is wrong with the methods, or that the birds have vastly different histories. Also, the last sentence of this paragraph is kind of a non sequitur, and it is unclear on what assumption of reaction the statement is based. Consistent reactions should not necessarily mean that they are “more prone to react”, just that they are consistent in reacting.

- *In response to both reviewers, we have erased this sentence as well as the entire paragraph –(which does not make much sense without this sentence). However, as a comment to the reviewers concern, we have been involved in many playback studies, and even though the study is well designed it does not guarantee that all individual animals react in a similar fashion (as they may be*

physiologically or cognitive 'busy' with different 'tasks'). The only other species (including birds, whales, seals and fish) that we have studied that show such predictable reactions to playbacks at very low level re. the hearing threshold is, as has now been pointed out, the harbour porpoise.

Line 292 The statement that “penguins probably react to novel sounds under water by fear rather than by curiosity” can’t be backed up by the current data, which are very specific in scope. This could be reworded in a way to highlight the topic of neophobic reactions to sound in penguins, but “probably” seems too much of a reach here.

- *OK; we have replaced 'probably' by 'might' to accommodate this, so the sentence now reads: 'The strong startle responses also indicate that penguins might react to novel sounds under water by fear rather than by curiosity.'*

Appendix B

Odense, 31st of January, 2020

Dear Editor,

Thank You for recommending our manuscript for publication.

Please find attached a revised version of the manuscript *Gentoo penguins (Pygoscelis papua) react to underwater sounds*, taking the revisions made by the Associate Editor into account. Our corrections are found below.

Please do not hesitate to contact us if there are any further issues.

Sincerely,

Kenneth Sørensen and coauthors

Gentoo penguins (*Pygoscelis papua*) react to underwater sounds

K. Sørensen^{1*}, C. Neumann¹, M. Dähne², K. A. Hansen,¹ M. Wahlberg¹

¹ Department of Biology, University of Southern Denmark,
Campusvej 55, 5230 Odense M, Denmark

² German Oceanographic Museum, Foundation, Katharinenberg 14-20, 18439 Stralsund, Germany

*Author of correspondence (kenneths@biology.sdu.dk)

**ABSTRACT**

Marine mammals and diving birds face several physiological challenges under water, affecting
their thermoregulation and locomotion as well as their sensory systems. Therefore, marine
mammals have modified ears for improved underwater hearing. Underwater hearing in birds has~~ve~~
been studied in a few species, but for the record-holding divers, such as penguins, there are no
detailed data. We played underwater noise bursts to gentoo penguins (*Pygoscelis papua*) in a large
tank at received sound pressure levels between 100 and 120 dB re 1 μ Pa rms. The penguins
showed a graded reaction to the noise bursts, ranging from no reactions at 100 dB to strong
reactions in more than 60% of the playbacks at 120 dB re 1 μ Pa. The responses were always
directed away from the sound source. The fact that penguins can detect and react to underwater
stimuli may indicate that they make use of sound stimuli for orientation and prey detection during
dives. Further, it suggests that penguins may be sensitive to anthropogenic noise, like many
species of marine mammals.

Keywords: bird hearing, penguins, bioacoustics, playback

**1. Introduction**

Marine mammals and diving birds are secondarily adapted to the aquatic environment. Their
anatomy and physiology, initially tuned for terrestrial life, have been modified to also function in
water. Some species can dive to depths of more than 500 m for more than an hour in pursuit of
their prey, which include fish, cephalopods, and crustaceans. Their bodies are streamlined to
reduce drag and preserve body heat while diving. Indeed, many species hunt for fish year-around
in cold waters while keeping their core body temperature intact [1].

In addition to these adaptations, some of the sensory systems of marine mammals and diving birds
have been adjusted to function well not only in air but also under water. In air, most mammals and
birds have acute visual and hearing abilities, and ~~especially mammals~~some also rely to a large part
on olfaction. When submerged, these senses function differently due to different properties of
water and air as propagation media for light, sound, and chemical stimuli. For example, the
difference in refraction index between air and water challenges the eyes' abilities to focus the
image on the retina [2]. The morphology of the eyes of whales and seals have been adapted to
function well both in air and under water [2, 3]. Some marine birds, such as penguins, also seem to
have anatomical adaptations to make their eyes function well in both media [4].

Besides vision, the sense of hearing of an air-adapted animal is also challenged under water. The
speed of sound is almost 4.5 times faster in water than in air. Together with the 800 times higher
density of water, this calls for anatomical changes in the detection system for it to function
optimally [5]. The high underwater speed of sound also challenges the animals' abilities to
determine the direction to a sound source. In air, directional hearing in the horizontal plane is
accomplished by measuring the time lag and intensity difference between the sound pulse received
at the two ears. Under water, neither of these cues function very well, as the higher speed of sound
results in shorter time lags as well as longer wavelengths, ~~and therefore~~ smaller intensity
differences between the two ears ~~and therefore less shielding of the skull for the signal arriving at~~
the two ears [5, 6].

Commented [KS1]: OK, thanks, we have rephrased this.

[revised manuscript text omitted]

figure 3. There was no ~~indication of change in~~ the response of the birds ~~changing~~ between
subsequent playback trials, which could have indicated ~~the birds getting habituated~~ habituation to
the sound emissions.

There was a clear graded response to the playback. This was obvious both when averaging the
response scores of all birds, and when analyzing the scores for individual birds. At exposure levels
of 100 and 105 dB re 1 μ Pa, only responses RS1 and RS2 were observed. At 110 and 115 dB re 1
μ Pa, there was one RS3, whereas at 120 dB re 1 μ Pa there was RS3 in more than 60% of the trials,
and RS2 and RS3 responses in more than 75% of the trials (figure 5). All eight control trials were
classified as RS0, ~~never showing any response of~~ with no response from the penguins.

The linear regression of the response and the intensity was significantly positive (ANOVA, $df = 4$
$p < 0.05$), and there was a significant effect from the sound level on the pooled RS0-1 responses
tested against RS2-3 responses (binominal test, $p < 0.05$).

4. Discussion

Commented [KS3]: Yes, you are right. We have corrected this.

All seven gentoo penguins reacted to the underwater noise pulses, and they did so in a graded
manner. Received levels below 115 dB re 1 μ Pa rms did not show any strong responses, whereas
the highest playback intensities of 115 and 120 dB re 1 μ Pa elicited many strong responses.
Combining the graded response to increased sound levels with the lack of responses for the control
trials, this clearly showed that the penguins reacted to underwater sounds (figures 4 and 5). The
stimulus contained frequencies between 0.2 and 6 kHz. This encompasses the known hearing
range of penguins in air [15] and makes it likely that the frequency range of underwater hearing is
similar. Judging by the gentoo penguins' fast swimming and curious behaviour, the extension of
the pause between playback trials, as well as the restriction on the maximum number of playbacks
270 per day, seemed to have been adequate to avoid any habituation to the signals. Indeed, there was
271 no sign in the data ~~on-of habituation-to-occur~~, and the penguins usually reverted to their playful
behaviour almost immediately after playback. In addition, noise from bubbles released from the
plumage while diving can potentially mask the bird's ability to hear the stimulus, but no bubbles
were observed at the time of any of the playbacks.

If we assume penguins have similar underwater hearing thresholds as the cormorant (about 70-75
277 dB re 1 μ Pa) [21], we expect the reaction threshold found in our study of some 115 dB re 1 μ Pa to
278 be 45-50 dB above the audiogram. This is surprising low. Comparing with humans, a sound
intensity 50 dB above the hearing threshold corresponds to speaking with a very soft voice and
therefore not usually causing any strong behavioural reactions. Also, behavioural reactions on
marine mammals are usually observed at received levels way beyond 100 dB relative the hearing
threshold [33], ~~with the exception of the only exemption seemingly being~~ the harbour porpoise
(*Phocoena phocoena*) sometimes showing reaction thresholds less than 40 dB relative the hearing
threshold [29]. The fact that penguins react strongly towards these rather weak sounds may either
indicate that they strongly disliked the noise burst, or that their underwater hearing threshold is
even lower than the cormorant's. ~~Any of these explanations would make sense only if we assume~~

[revised manuscript text omitted]

[8] Reichmuth, C., Holt, M.M., Mulsow, J., Sills, J.M. & Southall, B.L. 2013 Comparative assessment of amphibious
hearing in pinnipeds. *Journal of Comparative Physiology A* **199**, 491-507.
[9] Kastelein, R., Nieuwstraten, S., Staal, C., van Ligtenberg, C. & Versteegh, D. 1997 Low-frequency aerial hearing in a
Harbour Porpoise (*Phocoena phocoena*). (The biology of the Harbour Porpoise).
[10] Liebschner, A., Hanke, W., Miersch, L., Dehnhardt, G. & Sauerland, M. 2005 Sensitivity of a tucuxi (*Sotalia*
*fluviatilis guianensis*) to airborne sound. *The Journal of the Acoustical Society of America* **117**, 436-441.
[11] Ghoul, A. & Reichmuth, C. 2012 Sound production and reception in southern sea otters (*Enhydra lutris nereis*). In
*The effects of noise on aquatic life* (pp. 157-159, Springer).
[12] Gaspard, J.C., Bauer, G.B., Reep, R.L., Dziuk, K., Cardwell, A., Read, L. & Mann, D.A. 2012 Audiogram and
auditory critical ratios of two Florida manatees (*Trichechus manatus latirostris*). *Journal of Experimental Biology* **215**,
1442-1447.
[13] Christensen-Dalsgaard, J., Brandt, C., Willis, K.L., Christensen, C.B., Ketten, D., Edds-Walton, P., Fay, R.R.,
Madsen, P.T. & Carr, C.E. 2012 Specialization for underwater hearing by the tympanic middle ear of the turtle,
*Trachemys scripta elegans*. *Proceedings of the Royal Society of London B: Biological Sciences*, rspb20120290.
[14] Owen, M.A. & Bowles, A.E. 2011 In-air auditory psychophysics and the management of a threatened carnivore,
the polar bear (*Ursus maritimus*).
[15] Wever, E.G., Herman, P.N., Simmons, J.A. & Hertzler, D.R. 1969 Hearing in the blackfooted penguin, *Spheniscus*
*demersus*, as represented by the cochlear potentials. *Proceedings of the National Academy of Sciences* **63**, 676-680.
[16] Crowell, S.E., Wells-Berlin, A.M., Carr, C.E., Olsen, G.H., Therrien, R.E., Yannuzzi, S.E. & Ketten,
D.R.J.J.o.C.P.A. 2015 A comparison of auditory brainstem responses across diving bird species. **201**, 803-815.
[17] Mooney, T.A., Smith, A., Larsen, O.N., Hansen, K.A., Wahlberg, M. & Rasmussen, M.H. 2019 Field-based hearing
measurements of two seabird species. *Journal of Experimental Biology* **222**, jeb190710.
[18] Bradbury, J. & Vehrenkamp, S. 2011 Principles of Animal Communication. 3rd edn. New York: Sinaur Associates.
*Inc.[Google Scholar]*.
[19] Larsen, O.N., Dooling, R.J. & Michelsen, A. 2006 The role of pressure difference reception in the directional
hearing of budgerigars (*Melopsittacus undulatus*). *Journal of Comparative Physiology A* **192**, 1063-1072.

[20] Crowell, S.C. 2016 Measuring in-air and underwater hearing in seabirds. In *The Effects of Noise on Aquatic Life II*
(pp. 1155-1160, Springer.
[21] Hansen, K.A., Maxwell, A., Siebert, U., Larsen, O.N. & Wahlberg, M. 2017 Great cormorants (*Phalacrocorax*
*carbo*) can detect auditory cues while diving. *The Science of Nature* **104**, 45.
[22] Kooyman, G. & Kooyman, T. 1995 Diving behavior of emperor penguins nurturing chicks at Coulman Island,
Antarctica. *Condor*, 536-549.
[23] Howland, H.C. & Sivak, J.G. 1984 Penguin vision in air and water. *Vision research* **24**, 1905-1909.
[24] Strod, T., Izhaki, I., Arad, Z. & Katzir, G. 2008 Prey detection by great cormorant (*Phalacrocorax carbo sinensis*) in
clear and in turbid water. *Journal of Experimental Biology* **211**, 866-872.
[25] Zimmer, I., Wilson, R.P., Gilbert, C., Beaulieu, M., Ancel, A. & Plötz, J. 2008 Foraging movements of emperor
penguins at Pointe Géologie, Antarctica. *Polar Biology* **31**, 229-243.
[26] Ketten, D.R. 2008 Underwater ears and the physiology of impacts: Comparative liability for hearing loss in sea
turtles, birds, and mammals. *Bioacoustics* **17**, 312-315.
[27] Johansson, K. 2011 Impact of anthropogenic noise on fish behaviour and ecology.
[28] Wisniewska, D.M., Johnson, M., Teilmann, J., Siebert, U., Galatius, A., Dietz, R. & Madsen, P.T. 2018 High rates
of vessel noise disrupt foraging in wild harbour porpoises (*Phocoena phocoena*). *Proceedings of the Royal Society B:*
*Biological Sciences* **285**, 20172314.
[29] Tougaard, J., Wright, A.J. & Madsen, P.T. 2015 Cetacean noise criteria revisited in the light of proposed exposure
limits for harbour porpoises. *Marine pollution bulletin* **90**, 196-208.
[30] Frantzis, A. 1998 Does acoustic testing strand whales? *Nature* **392**, 29.
[31] Madsen, P.T. & Wahlberg, M. 2007 Recording and quantification of ultrasonic echolocation clicks from free-ranging
toothed whales. *Deep Sea Research Part I: Oceanographic Research Papers* **54**, 1421-1444.
[32] Sørensen, K., Neumann, C., Dähne, M., Hansen, K.A. & Wahlberg, M. 2019 Gentoo penguins (*Pygoscelis papua*)
react to underwater sounds. (Dryad Digital Repository. (<https://doi.org/10.5061/dryad.cc2fqz62v>)).
[33] Southall, B.L., Bowles, A.E., Ellison, W.T., Finneran, J.J., Gentry, R.L., Greene Jr, C.R., Kastak, D., Ketten, D.R.,
Miller, J.H. & Nachtigall, P.E. 2008 Marine mammal noise-exposure criteria: initial scientific recommendations.
*Bioacoustics* **17**, 273-275.
[34] Ketten, D. 1994 Functional analyses of whale ears: adaptations for underwater hearing. In *OCEANS-*
*CONFERENCE*- (pp. I-264, INSTITUTE OF ELECTRICAL & ELECTRONICS ENGINEERS.
[35] Frost, P., Shaughnessy, P., Semmelink, A., Sketch, M. & Siegfried, W. 1975 Response of Jackass Penguins to
Killer Whale vocalizations. *South African Journal of Science* **71**, 157-158.
[36] Takahashi, A., Sato, K., Naito, Y., Dunn, M., Trathan, P. & Croxall, J. 2004 Penguin-mounted cameras glimpse
underwater group behaviour. *Proceedings of the Royal Society of London. Series B: Biological Sciences* **271**, S281-
S282.
[37] Pichegru, L., Nyengera, R., McInnes, A.M. & Pistorius, P. 2017 Avoidance of seismic survey activities by penguins.
*Scientific reports* **7**, 16305.

**Tables**

**Table 1.** Definitions of Response Scores 0-3.

score	name	definition
RS 0	no response	no head movement or change in direction and speed
RS 1	minor behavioural response	45-90° change of head movement
RS 2	clear behavioural response	45-90° change of swim direction and change of speed
RS 3	startle response:	>90° change of swim direction and change of speed

**Table 2.** Number of playback trials for each of the seven gentoo penguins.

sound level (dB re 1 μ Pa)	individual						
	A	B	C	D	E	F	G
100	0	1	3	1	0	2	1
105	0	2	3	2	1	0	0
110	1	2	1	2	0	1	1
115	0	1	3	1	1	0	2
120	1	1	1	0	1	2	2

**Figure captions**

**Figure 1.** Drawing of the penguin enclosure in Odense Zoo. Playback setup is pointing towards the pool center

(encircled with red). A: deck area, B: pool, C: public viewing area, D: experimenter's position.

**Figure 2.** Oscillogram and power spectral density plot (solid line) of received 120 dB re 1 μ Pa signal and ambient

noise (stippled line), measured 1 m in front of the loudspeaker. Dotted line is the recording system self-noise.

**Figure 3.** Video frame sequences of a trial with no response (RS 0) at 100 dB re 1 μ Pa, a minor behavioural response

(RS 1) at 110 dB re 1 μ Pa, a clear behavioural response (RS 2) at 120 dB re 1 μ Pa and a startle response (RS 3) at

120 dB re 1 μ Pa. The sequences show the penguin 1 s before until 1 s after playback. The transducer (encircled with

red) is at the top of each frame directed horizontally at 1 m depth.

**Figure 4.** Response to playback by seven gentoo penguins (each individual indicated with different colours and

shapes). Overlapping data points have been jittered for clarity.

**Figure 5.** Stacked bar chart showing response scores (RS 0-3) for different stimulus levels and controls. Each colour

represents different response scores.